# Effect of an enhanced public health contact tracing intervention on the secondary transmission of SARS-CoV-2 in educational settings: The four-way decomposition analysis

Olivera Djuric[1,2]*, Elisabetta Larosa[3], Mariateresa Cassinadri[3], Silvia Cilloni[3], Eufemia Bisaccia[3], Davide Pepe[3], Laura Bonvicini[1], Massimo Vicentini[1], Francesco Venturelli[1], Paolo Giorgi Rossi[1], Patrizio Pezzotti[4], Alberto Mateo Urdiales[4], Emanuela Bedeschi[3], The Reggio Emilia Covid-19 Working Group

[1]Epidemiology Unit, Azienda Unità Sanitaria Locale – IRCCS di Reggio Emilia, Reggio Emilia, Italy; [2]Centre for Environmental, Nutritional and Genetic Epidemiology (CREAGEN), University of Modena and Reggio Emilia, Modena, Italy; [3]Public Health Unit, Azienda Unità Sanitaria Locale-IRCCS di Reggio Emilia, Reggio Emilia, Italy; [4]Department of Infectious Diseases, Istituto Superiore di Sanità, Rome, Italy

*For correspondence: olivera.duric@ausl.re.it

## Abstract

**Background:** The aim of our study was to test the hypothesis that the community contact tracing strategy of testing contacts in households immediately instead of at the end of quarantine had an impact on the transmission of SARS-CoV-2 in schools in Reggio Emilia Province.

**Methods:** We analysed surveillance data on notification of COVID-19 cases in schools between 1 September 2020 and 4 April 2021. We have applied a mediation analysis that allows for interaction between the intervention (before/after period) and the mediator.

**Results:** Median tracing delay decreased from 7 to 3.1 days and the percentage of the known infection source increased from 34–54.8% (incident rate ratio-IRR 1.61 1.40–1.86). Implementation of prompt contact tracing was associated with a 10% decrease in the number of secondary cases (excess relative risk –0.1 95% CI –0.35–0.15). Knowing the source of infection of the index case led to a decrease in secondary transmission (IRR 0.75 95% CI 0.63–0.91) while the decrease in tracing delay was associated with decreased risk of secondary cases (1/IRR 0.97 95% CI 0.94–1.01 per one day of delay). The direct effect of the intervention accounted for the 29% decrease in the number of secondary cases (excess relative risk –0.29 95%–0.61 to 0.03).

**Conclusions:** Prompt contact testing in the community reduces the time of contact tracing and increases the ability to identify the source of infection in school outbreaks. Although there are strong reasons for thinking it is a causal link, observed differences can be also due to differences in the force of infection and to other control measures put in place.

**Funding:** This project was carried out with the technical and financial support of the Italian Ministry of Health – CCM 2020 and Ricerca Corrente Annual Program 2023.

## Editor's evaluation

This study provides a useful assessment of the effect of testing contacts of cases in school classes when identified, rather than at the end of quarantine, on both the number of secondary infections and other outcomes including tracing delay and identification of the possible source of infection. The authors find solid evidence that the intervention led to a decrease in tracing delay, an increase in the known source of infection of index cases and that knowing the source of the index case reduced secondary transmission. The novel application of the mediation analysis will be of interest to infectious disease epidemiologists and the findings will be of interest to those planning for future pandemics.

## Introduction

Closure of educational institutions was one of the non-pharmacological infection control measures often adopted during the pandemic of SARS-CoV-2, mostly based on the temporal coincidence between schools reopening and COVID-19 outbreaks in some countries and the concern regarding potential school-to-home transmissions of the virus from students to more susceptible family members.

Evidence of SARS-CoV-2 transmission in educational settings indicates not only that schools opening and closing have a small impact on the increase or decrease of SARS-CoV-2 rates in the population, but that transmission is even lower in schools than that in the general population (*Winje et al., 2022*; *Gandini et al., 2021*; *Viner et al., 2022*). However, the risk of in-school SARS-CoV-2 transmission is still considered high, making prevention measures vital to restoring in-person learning (*European Centre for Disease Prevention and Control, 2020a*; *European Centre for Disease Prevention and Control, 2020b*). The control of infection in school-age children became even more critical after the introduction of mass vaccination, which reduced transmission between adults, and with the spread of the Omicron variant, which has much higher transmissibility in indoor settings.

Timely reporting of COVID-19 cases to the health authorities and case investigation, followed by timely testing, contact tracing, and isolation, remain crucial to allow safe resumption of in-presence activities. Contact tracing practices have been subject to changes over time along with emerging evidence and the introduction of the vaccine. In the operational document from September 2021, the Centers for Disease Control and Prevention (CDC) recommended that people get tested at least after five days from close contact with a person with COVID-19 *Centers for Disease Control and Prevention, 2021*, while the European Centre for Disease Prevention and Control (ECDC) recommended testing all high-risk exposure contacts, whether vaccinated or not, as soon as possible after they have been identified to allow for further contact tracing (*European Centre for Disease Prevention and Control, 2020b*). Regardless of this, it has always been acknowledged that isolation of contacts is effective if initiated shortly after confirmation of the index case since the delay in isolation of contacts has a major impact on the transmission of the virus (*Kretzschmar et al., 2020*).

Enhanced contact tracing, such as backward contact tracing, has also been recommended to facilitate the identification of the primary case, also called 'source' or 'original' case from which an index case acquired his/her infection (*European Centre for Disease Prevention and Control, 2020b*; *Centers for Disease Control and Prevention, 2021*). The rationale behind this recommendation is to stop chain transmission that originates from this relatively small proportion of primary cases usually responsible for a large proportion of transmission. By extending the contact tracing window or performing source investigation, BCT aims to identify asymptomatic cases that are the actual source of newly detected (index) cases. Modeling studies show that primary cases generate 3–10 times more infections than a randomly chosen case (*Endo et al., 2020*). These cases would not have otherwise been identified and, in the case of educational settings, would not have been linked to school investigation. Given that BCT tends to 'catch' infection sources at the end of their infectious period, it is highly susceptible to testing and contact tracing delays, therefore, it is meaningful only in the presence of prompt tracing of contacts (*Raymenants et al., 2022*).

Starting from 27 November 2020, the local health authority of Reggio Emilia, Italy, improved contact tracing protocols by introducing prompt molecular tests for all contacts, whether symptomatic or asymptomatic, at the beginning of quarantine (test to trace), with the aim to identify all possible sources of infection in asymptomatic contacts. Before the intervention, contacts of index cases were only tested at the end of the isolation period (test to release). In this way, primary (asymptomatic) cases were not diagnosed or were diagnosed very late in their infection course and, given that they

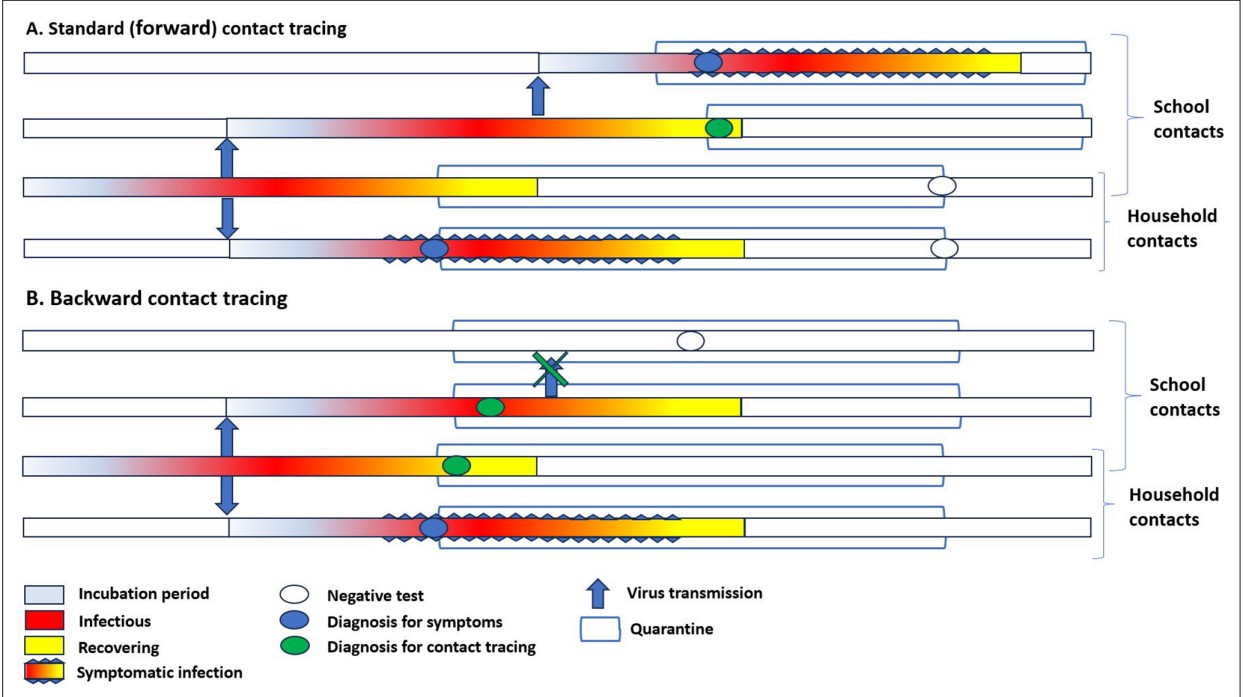

**Figure 1.** Simplified timeline of transmission in household and among school contacts in the presence of standard contact tracing and backward contact tracing. (**A**) In **standard contact** tracing, all close contacts were quarantined after identifying a case in the community. Contacts were only tested at the end of the quarantine or if symptomatic. Only for school contacts, immediate testing of all classmates was performed; if one or more classmates resulted positive, the whole class was quarantined. (**B**) In **backward contact tracing**, close contacts were also immediately tested, independently from the presence of symptoms. The tracing and quarantine policy in schools was similar. In the proposed example, after the diagnosis of a symptomatic household member, backward tracing would identify an asymptomatic child, thus allowing the extension of investigation to his school contacts and eventually stopping secondary transmission in the class.

were not attending school since they were isolated, they were not indicated as a school contact until one of the school contacts become symptomatic (*Figure 1* and *Figure 2*). Given that a large part of the infections in students is asymptomatic or paucisymptomatic, they are often identified when an adult in the same household presents symptoms. Prompt testing of all contacts in community allows the timely identification of positive children/teachers who may be primary cases in school outbreaks, thus, allowing a prompt investigation in the school setting to start.

This study aimed to estimate the impact of changing contact tracing intervention from testing contacts at the end of quarantine to testing contacts immediately, on the secondary transmission of SARS-CoV-2 in educational settings in Reggio Emilia Province. To better understand the mechanism of the possible impact that the intervention has on secondary transmission, we assessed whether this association is mediated by two process indicators, tracing delay and effective tracing, measured as known sources of infection of the index case and proportion of asymptomatic index cases, which were the actual target of the intervention, bearing in mind limits of the before-and-after design of this study conducted in a period when several changes could confound the results.

## Methods
### Design and setting

In the present study, population-based surveillance data were analysed including 1604 consecutive positive cases confirmed with RT-PCR for SARS-COV-2 infection between 1 September 2020 and 4 April 2021 in Reggio Emilia Province that led to an epidemiological investigation among children and adolescents (0–19 years old) or school staff in 1884 classes who may have been exposed or in contact with positive cases at school.

In Reggio Emilia Province (531,751 inhabitants, Emilia Romagna, Northern Italy) there are approximately 95,000 inhabitants from 6 months-olds to 19-year-olds attending infant-toddler centres (ages

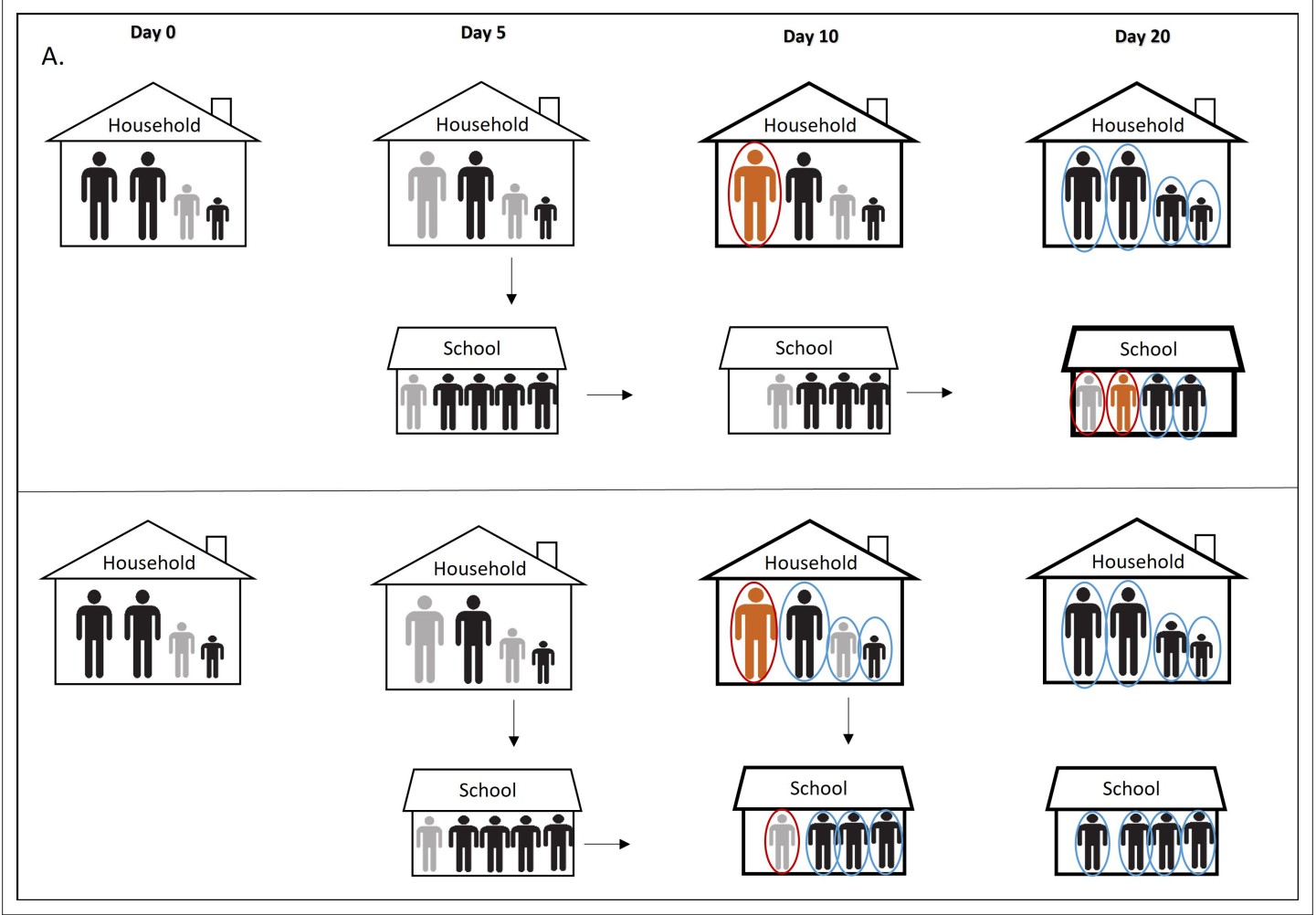

**Figure 2.** Simplified illustration of the pre and post intervention scenarios. In panel A we report the scenario without prompt contact testing in community and its effect on the SARS-CoV-2 transmission in educational setting. Day 0: One of the children in a household became infected (primary case) but asymptomatic (gray). Day 5: One parent and one classmate became infected, also asymptomatic (gray). Day 10: The infected parent became symptomatic (orange), tested positive (red circle), and considered an index case of the household. Entire family is quarantined (bold line) but not tested immediately. Meanwhile, the primary case transmits infection further to two other classmates. Classmates of the primary case are not tested because they are not identified as school contacts due to late testing of the household contacts. Day 20: Family members of the index case are tested at the end of the quarantine. One positive classmate of the primary case became symptomatic, tested positive, and considered an index case in the school cluster given that the classmates were not considered contacts of the primary case since he was already isolated. Other classmates are tested only when an index case occurs. Panel B illustrates the scenario with prompt contact testing in community. Day 10: The infected parent became symptomatic, tested positive, and entire family was quarantined and tested at the beginning of quarantine. Primary case is identified promptly, his classmates are identified as contacts, tested, and isolated preventing further transmission of the virus.

0–3), preschools (ages 3–5), primary schools (ages 6–10), middle schools (ages 11–13), and high schools (ages 14–19), and about 12,000 teachers/school staff members.

During the study period, there were two peaks of infections: in November 2020 and in February/ March 2021 (*Figure 3*; *Istituto Superiore di Sanità, 2021*). After the school reopening on 1 September 2020 for preschool and remedial courses and, on 15 September 2020, for the regular school year, in-class learning was in place until 26 October 2020 when policies to reduce crowding especially in high schools were introduced (reducing the in-class time by 50–75%) as were several short closures in the periods of highest incidence. In addition, because of the high circulation of the virus, the Christmas school holidays were extended to the second week of January (from 20 December to 11/15 January). Another lockdown led to the closing of schools on 3 March 2021. Only infant-toddler centres and preschools, schools that require laboratory work, and schools with pupils with disabilities or special needs continued in-presence didactic activities.

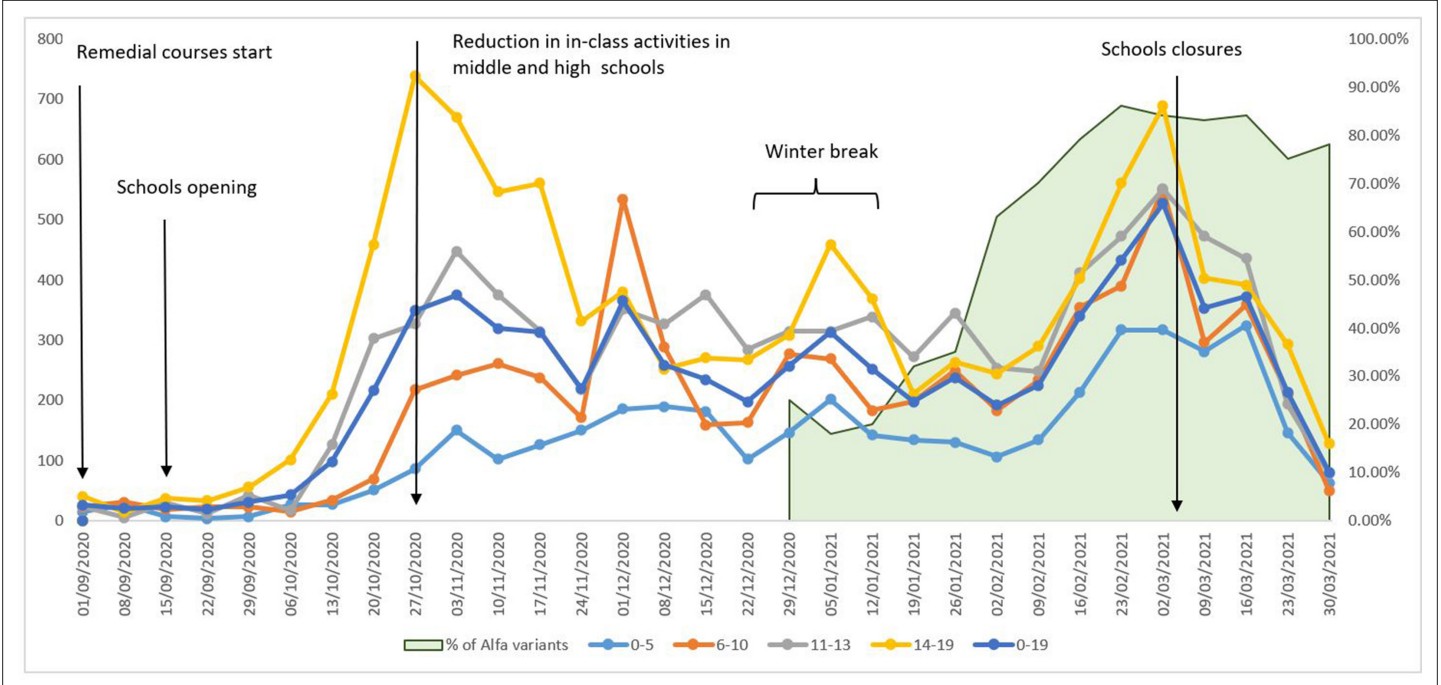

**Figure 3.** Weekly notification rates of new COVID-19 cases per 100,000 inhabitants, ages 0–19, by age class, Reggio Emilia Province, 1 September 2020 – 4 April 2021. The graph also reports the main changes in school opening and school closures and the proportion of Alpha variants (green area) among sequenced cases reported by the Italian National Institute of Health.

Infection control measures in place during the study period were previously described in detail *Larosa et al., 2020*; *Djuric et al., 2022*; *Regione et al., 2020*.

## Intervention

Starting from 27 November 2020, the local health authority improved contact tracing protocols and introduced immediate molecular tests for all contacts, whether symptomatic or asymptomatic, at the beginning of quarantine, with the aim to identify all possible sources of infection in asymptomatic contacts and facilitate backward tracing (*Djuric et al., 2022*). This strategy was applied to all contacts, independently from the setting of infection, including all household members of sporadic cases, and particular attention was given to testing of children and adolescents because they were most commonly asymptomatic. This strategy was explicitly thought to correctly identify in a timely manner the contacts of asymptomatic cases before they started the quarantine. Testing only at the end of quarantine guarantees a safe return to the community of contacts and to identify secondary transmission in the cluster, but, by definition, assumes that the asymptomatic cases are secondary cases and became infectious during the quarantine and thus could not have contacts.

## Outcome and variables of interest

The main outcome was the number of secondary cases per class; we preferred to use the absolute number instead of the attack rate, because we were interested in assessing whether the intervention limited the number of secondary cases and not the probability of being infected given that an exposure occurred. Three process indicators of contact tracing performance were considered. The first one, tracing delay, was calculated as the time from the swab positivity of the index case to the date on which the swab for (the majority of) classmates was scheduled. The second indicator was the proportion of index cases who had close contact with a known COVID-19 case in the ten days before the onset of symptoms or diagnosis. This indicator, called 'the known source of infection of the index case,' is a proxy of backward contact tracing success, which should reflect the extent to which school index cases were tested and linked to the school investigation because of a known contact with a positive person. Finally, the third indicator was the proportion of asymptomatic index cases. This indicator is also a proxy of backward tracing, because in the absence of screening, asymptomatic cases

are mostly identified during contact tracing and to become an index case of school investigation this testing should not be done at the end of quarantine. We also reported testing delay, i.e., the delay in the diagnosis of the index case, defined as the number of days between symptom onset and the date of swab positivity, but this indicator is expected to only be marginally influenced by contact tracing strategies.

## Definitions and assumptions

The first case that tested positive (considering the date on which the swab was done) per class was considered an index case. If more than one case in a class tested positive on the same day, the one with the earliest symptom onset was considered the index case. The same class can be included more than once in the analysis because it may have been involved in more than one investigation during the study period.

When more than one class was included in a between-class transmission, index cases belonging to different classes had shared exposures, or there was a single index case for more than one class (usually, but not only, when the index case was a teacher), this was considered a multi-class cluster.

The overall attack rate was calculated by dividing the number of cases by the population at risk; i.e., classmates, teachers/staff who had had close contact with the index case in a period starting from 48 hr before symptom onset of the symptomatic index case and, for asymptomatic cases, 48 hr before diagnosis.

If a classmate was already in isolation prior to symptom onset or swab positivity of the index case, due to contact with a positive person or re-entry from abroad, he/she was excluded from the denominator. Any student or staff who refused to perform a swab was excluded from the denominator.

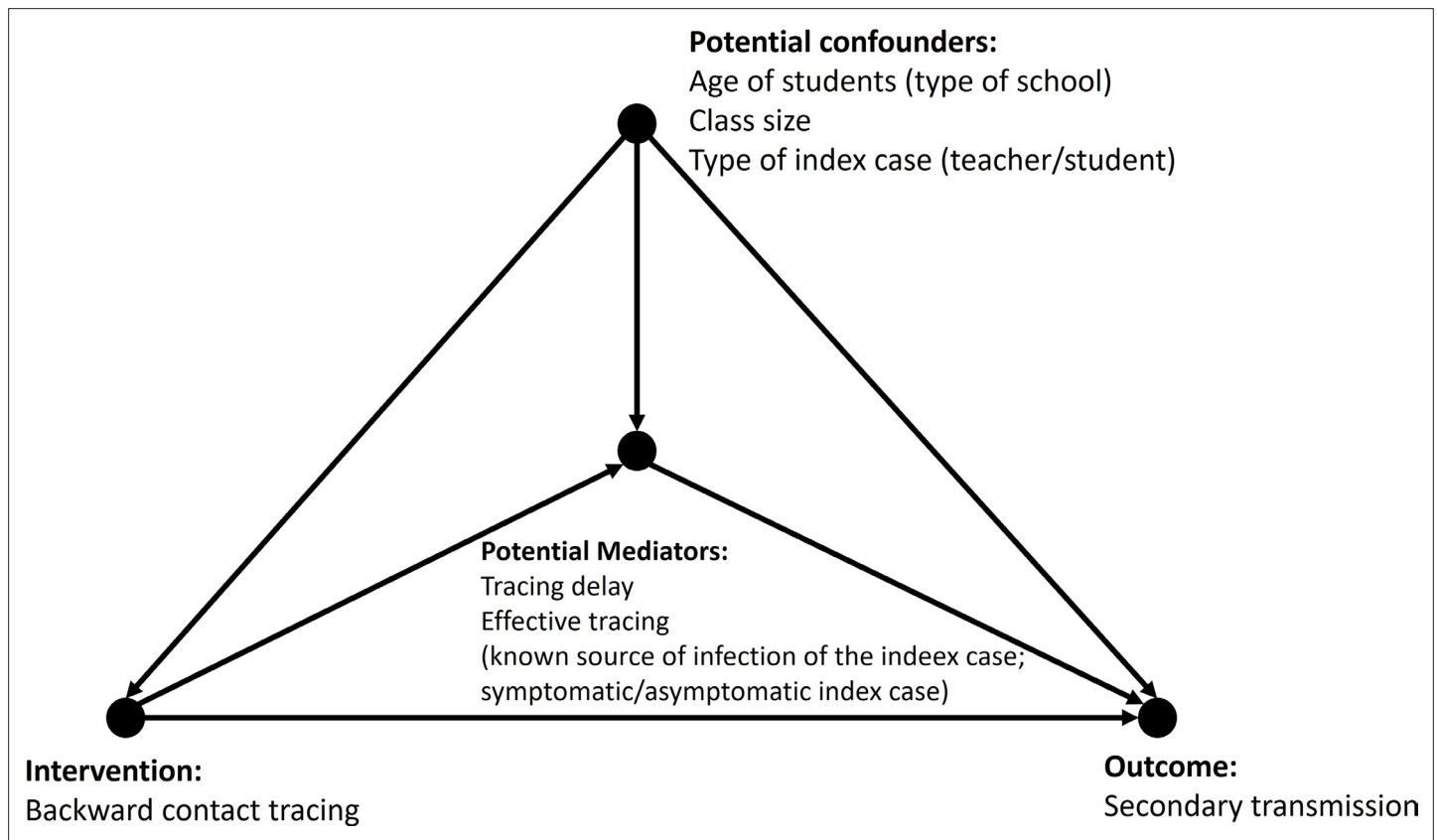

**Figure 4.** Directed acyclic graph for the association of intervention and number of secondary cases, mediated by contact tracing delay and known contact of the index case, adjusted for the type of school, type of index case, and class size.

## Data sources

Following the identification and notification of a COVID-19 case, qualified Public Health Department (PHD) personnel performed a detailed field investigation and managed the index case and identified contacts according to the regional recommendations and control measures in place. Comprehensive surveillance data containing information on index cases, contacts, school and class characteristics, swabs performed, secondary cases, and measures undertaken, were collected by PHD, and stored in electronic forms. Each case and cluster were re-abstracted by a study investigator and checked for consistency and plausibility. Missing data were imputed from the COVID-19 Surveillance Registry software and a de-identified research database was constructed for the analysis.

## Statistical analysis

During the study period, many factors that could influence secondary transmission in schools occurred, including changes in overall incidence, changes in in-school and out-of-school (especially transport and leisure-time activities) control measures, time of in-person and distance teaching, and the spread of the Alpha variant. Therefore, simply measuring the outcome before and after the intervention would be surely biased and would not allow any causal inference.

To test the hypothesis that the contact tracing strategy in the community had an impact on the secondary case transmission in schools, we defined a direct acyclic graph identifying the possible causal pathways, including possible mediators and possible confounders (*Figure 4*).

First, we assessed if the possible known confounders were associated with the intervention (i.e. were they differently distributed in the before and after periods), and with the outcome (i.e. the number of secondary cases). Then we assessed if the introduction of the new strategy actually changed the tracing process analysing the trend of the timeliness of testing and effectiveness of tracing (measured as the proportion of index cases with a known source of infection and proportion of symptomatic index cases) during the study period. The class was the statistical unit for analyses. Median tracing delay and the proportion of index cases with the known source of infection were compared before and after the implementation of the intervention (27 November 2020). Second, we tested the association between the three process indicators that were the direct target of the new tracing strategy and the final health outcome (number of secondary cases). Three negative binomial regression models were constructed with the number of secondary cases per class as outcome and intervention indicators as exposures. Models were adjusted for types of school (infant-toddler centre, primary school, middle school, high school, other educational services), class size (<21 and ≥21 pupils), and types of the index case (student vs. teacher). Given that tracing delay and known contact were strongly negatively correlated ($r=–0.76$), their effects were analysed separately. With a similar model, we also measured the association between the intervention and the outcome.

Lastly, a novel effect decomposition method was used in a subset of pre-Alpha variant (before 31 December 2020) classes to test whether one of the two process indicators mediated the association between intervention and the number of secondary cases. This method was chosen because we assumed that there might be an association between exposure variables (before and after public health intervention) and mediator variables (process indicators). Assumptions about mediation analysis on unconfounded associations between variables were tested by performing a set of abovementioned analyses and following and based on the direct acyclic graph. The total effect of the intervention on the number of secondary cases is expressed as the excess relative risk (ERR); i.e., an incidence risk ratio (IRR) from the negative binomial regression minus one. In the presence of an intervention-mediator interaction, ERR is decomposed into four components: controlled directed effect (CDE) due to intervention only, at a fixed level of the mediator; pure indirect effect (PIE) due to mediation only; reference interaction (IntRef) due to interaction only; and mediated interaction (IntMed) due to mediation and interaction (*VanderWeele, 2014*). *Supplementary file 1* reports a plain language definition of the mediation analysis definitions. Given that classes could not be randomly assigned to the intervention and control group, and that the period before the intervention was used for the comparison, we assume that there might be a substantial interaction between the period before and after the intervention and two mediators. Stata's 'Med4way' command was used to estimate mediation and interaction effects simultaneously (*Discacciati et al., 2019*). Incidence rate ratios with a 95% confidence interval (CI) were reported and used for hypothesis testing. The Stata code used is

provided in the *Supplementary file 2*. All analyses in this study were conducted using STATA 13.0 SE (Stata Corporation, Texas, TX).

## Results

### Description of investigated classes and secondary transmission

We investigated 1884 classes overall, 1882 in which at least one case/contact was recorded, and two classes where screening was done due to out-of-school contact with an index case from another class. One thousand seven hundred and five secondary cases (1047 students and 658 teachers/staff) were identified among 43,214 tested contacts linked to 1604 index cases, resulting in an overall secondary attack rate of 3.9% (95%CI 3.8–4.1).

The median number of secondary cases per class was 1 (IQR 1–3); 2 before, and 1 after the intervention (test of equal medians p=0.092) (*Table 1*). The proportion of classes where secondary transmission occurred was overall 38.6%; 37.4% before and 39.0% after the intervention.

The number of symptomatic index cases decreased in the period after intervention from 85.3–80%. There were no changes in the number of classes that made up part of a multi-class cluster, as well as in the type of index case.

Secondary transmission was associated with the type of index case; it was lower among teachers than among students (IRR 0.75 95% CI 0.61–0.92) (*Table 2*).

We also tested the association between class or index case characteristics and the process indicators (*Table 3*). There was no difference in the number of index cases with known sources between types of school and class size. Percentage of known sources of infection was higher when the index case was a student compared to teachers (56.3% vs 26.7%). Median tracing delay was 3 days in all types of schools and index cases. There was more symptomatic index cases in infant-toddler centres and high schools than in primary schools and other educational services.

### Association between intervention and process indicators

Overall median tracing delay was 3 days (IQR 2–5), decreasing from 7 (IQR 5–10) in the period before intervention to 3.1 (IQR 2–4) days in the period after intervention (*Table 1*). The testing delay also decreased from 5 to 4 days following the implementation of the intervention. The percentage of index cases with a known source of infection was 49.3%, and it increased from 34% in November to 54.8% in the period after intervention. The number of index cases that were part of a household outbreak increased from 66.4% before the intervention to 80.3% after the intervention. Weekly average contact tracing delay decreased while the percentage of known sources of infection increased in the period after intervention implementation (*Figure 5*).

### Association between process indicators and outcome

Results of negative binomial regression covering the entire period show that both known sources of infection (IRR 0.75 95% CI 0.63–0.91) and decrease in tracing delay (1/IRR 0.97 95% CI 0.94–1.01 for each day of avoided delay) were associated with the decrease of the number of secondary cases (*Table 4*). Sensitivity analyses restricted to the period before the spread of the Alpha variant showed similar results (*Table 4*).

### Mediation analysis

Only the known source of infection of the index case was associated with the outcome (number of secondary cases) in multivariable analysis and it was, therefore, tested for the mediation and interaction in the four-way decomposition method.

Implementation of prompt contact tracing was associated with a 10% decrease in the number of secondary cases (excess relative risk –0.1 95% CI –0.35–0.15) (*Table 5*). The direct effect of the intervention accounted for the large part of the excess in risk (excess relative risk –0.29 95%–0.61– 0.03), leading to the 29% decrease in the number of secondary cases if the source of infection of the index case is known. Interaction only accounted for the other large part of the excess risk (excess relative risk 0.35 95% 0.03–0.68); knowing the source of infection of the index case in the period before the intervention when tracing delay was high, would increase the risk of secondary cases by 35%. However, we found evidence of mediated interaction that had a negative effect on the secondary transmission

**Table 1.** Characteristics of 1884 classes and 1604 index cases for which a school contact with COVID-19 cases was suspected, before, and after the intervention.

| | n (%) | Before intervention n=490 | After intervention n=1394 |
|---|---|---|---|
| **Classes (n=1884)** | | | |
| Type of school | | | |
| Infant-toddler centre | 350 (18.5) | 107 (21.8) | 243 (17.4) |
| Primary school | 540 (28.7) | 125 (25.5) | 415 (29.8) |
| Middle school | 496 (26.3) | 128 (26.1) | 368 (26.4) |
| High school | 478 (25.4) | 129 (26.3) | 349 (25.0) |
| Other educational services | 20 (1.1) | 1 (0.2) | 19 (1.4) |
| Calendar period | | | |
| September/October | 248 (13.1) | | |
| November | 263 (13.9) | | |
| December | 316 (16.8) | | |
| January | 265 (14.1) | | |
| February | 523 (27.8) | | |
| March/April | 269 (14.3) | | |
| Class size | | | |
| <21 | 862 (45.7) | 191 (39.0) | 671 (48.1) |
| ≥21 | 1011 (53.7) | 293 (59.8) | 718 (51.5) |
| Missing | 11 (0.6) | 6 (1.2) | 5 (0.4) |
| Secondary transmission | | | |
| No | 1157 (61.4) | 307 (62.6) | 850 (61.0) |
| Yes | 727 (38.6) | 183 (37.4) | 544 (39.0) |
| Number of secondary cases* | 1 (1-3) | 2 (1-3) | 1 (1-3) |
| Mean attack rate | 0.1 (0.04–0.12) | 0.1 (0.04–0.12) | 0.1 (0.04–0.12) |
| Part of a school cluster | | | |
| No | 1 367 (72.6) | 368 (75.1) | 999 (71.7) |
| Yes | 517 (27.4) | 122 (24.9) | 395 (28.3) |
| Tracing delay* | 3 (2-5) | 7 (5-10) | 3 (2-4) |
| Testing delay* | 4 (2-8) | 5 (3-8) | 4 (2-7) |
| **Index cases (n=1604)** | | n=429 | n=1,175 |
| Type of index case | | | |
| Student | 1213 (75.6) | 321 (74.8) | 892 (75·9) |
| Teacher | 391 (24.4) | 108 (25.2) | 283 (24.1) |
| Index case symptomatic | | | |
| No | 298 (18.6) | 63 (14.7) | 235 (20) |
| Yes | 1306 (81.4) | 366 (85.3) | 940 (80) |
| Potential source of infection | | | |
| Unknown | 814 (50.7) | 283 (66·0) | 531 (45.2) |

*Table 1 continued on next page*

*Table 1 continued*

|  | n (%) | Before intervention n=490 | After intervention n=1394 |
|---|---|---|---|
| Known | 790 (49.3) | 146 (34·0) | 644 (54.8) |
| Type of source |  |  |  |
| Household outbreak | 614 (77.7) | 97 (66.4) | 517 (80.3) |
| Social contact | 26 (3.3) | 7 (4.8) | 19 (2.9) |
| Sport contact | 18 (2.3) | 7 (4.8) | 11 (1.7) |
| Unidentifiable contact | 132 (16.7) | 35 (24.0) | 97 (15.1) |

*Median (IQR), calculated only in classes with secondary transmission.

(excess relative risk –0.14 95% CI –0.28–0.01). The known source of infection of the index case alone accounted for only a small percent of the reduction of excess risk (excess relative risk –0.02 95% –0.10–0.07).

## Discussion

We found that both process indicators used to evaluate the contact tracing intervention (tracing delay and known source of infection of the index case) improved after implementation of the public health intervention while the median number of secondary cases decreased, despite the higher daily absolute number of classes investigated in the period after the intervention. However, only the known source of infection of the index case evinced an association with a decrease in secondary transmission in school classes.

Our findings are consistent with those of modeling studies reporting that contact tracing efficacy decreases sharply with increasing delays between symptom onset and tracing and with a lower fraction of symptomatic infections being tested, fewer cases ascertained by contact tracing, and increasing transmission before symptom onset (*Kretzschmar et al., 2020*; *Gardner and Kilpatrick, 2021*;

**Table 2.** Association between class or index case characteristics (potential confounders) and number of secondary cases.

|  | Number of classes with secondary transmission | Number of secondary cases | IRR* (95% CI) |
|---|---|---|---|
| Classes (n=1884) | n=727 | n=1706 |  |
| Type of school |  |  |  |
| Infant-toddler centre | 131 (18.2) | 349 | ref |
| Primary school | 217 (29.8) | 553 | 1.03 (0.80–1.31) |
| Middle school | 172 (23.7) | 386 | 0.78 (0.60–1.01) |
| High school | 202 (27.8) | 409 | 0.86 (0.66–1.11) |
| Other educational services | 5 (0.7) | 9 | 0.45 (0.17–1.18) |
| Class size |  |  |  |
| <21 | 316 (43.5) | 751 | ref |
| ≥21 | 411 (56.5) | 955 | 1.08 (0.91–1.29) |
| Index cases (n=1604) | n=640 |  |  |
| Type of index case |  |  |  |
| Student | 477 (74.5) | 1047 | ref |
| Teacher | 163 (25.5) | 658 | 0.75 (0.61–0.92) |
| Screening | 0 | 1 | na |

*Relative risks are computed with negative binomial models with the count of secondary cases as a dependent variable.

**Table 3.** Association between class or index case characteristics and the process indicators (potential mediators).

| | Total n | Known source of infection of the index case n (%)* | Index case symptomatic n (%)* | Tracing delay Median (IQR) |
|---|---|---|---|---|
| **Classes (n=1884)** | | | | |
| **Type of school** | | | | |
| Infant-toddler centre | 350 | 157 (44.9) | 281 (80.3) | 3 (2-5) |
| Primary school | 540 | 255 (47.2) | 355 (65.7) | 3 (2-5) |
| Middle school | 496 | 206 (41.5) | 293 (59.1) | 3 (2-6) |
| High school | 478 | 216 (45.2) | 360 (75.3) | 3 (2-6) |
| Other educational services | 20 | 7 (35.0) | 10 (50.0) | 3 (1.5–5) |
| P value† | | 0.378 | 0.001 | 0.147 |
| **Class size** | | | | |
| <21 | 862 | 375 (41.4) | 587 (81.6) | 3 (2-5) |
| ≥21 | 1011 | 461 (45.6) | 707 (69.9) | 3 (2-6) |
| Missing | 11 | 5 (45.5) | 5 (45.5) | 6 (3-7) |
| p value† | | 0.661 | 0.782 | 0.367 |
| **Index cases (n=1604)** | | | | |
| **Type of index case** | | | | |
| Student | 1213 | 683 (56.3) | 957 (78.9) | 3 (2-5) |
| Teacher | 391 | 104 (26.7) | 342 (87.5) | 3 (2-6) |
| p value† | | <0.001 | <0.001 | 0.486 |

*Values are numbers with row percentages.
†Kruskal-Wallis test.

*Bradshaw et al., 2021*; *Hellewell et al., 2020*). Moreover, our previous modelling study showed that identifying positive cases within 5 days after exposure to their infector could reduce by 30% onward transmission at schools (*Molina Grané et al., 2023*). Observational studies also demonstrated that various improvements in contact tracing (*Malheiro et al., 2020*) can reduce the secondary transmission or even mortality in the community (*Vecino-Ortiz et al., 2021*). A few open-label and field trials conducted with intention to minimize confounding showed that daily testing and twice-a-week testing strategies are effective in limiting the secondary transmission while reducing the loss of in-person school days (*Young et al., 2021*; *Harris-McCoy et al., 2021*).

Our results suggest that there is a modest association between the intervention and the number of secondary cases. It has been shown that the effectiveness of contact tracing highly depends on the number of cases being traced, i.e., it decreases when the burden of new cases is too high for the tracing capacity of the health services (*Gardner and Kilpatrick, 2021*). In fact, BCT is more effective when community transmission is low to moderate (*Ontario Agency for Health Protection and Promotion (Public Health Ontario), 2021*). Similarly, increased new cases burden and high transmission during the winter months in our study could be factors that might have minimised the true effect of the intervention.

Interestingly, tracing delay was not associated with the decrease in the secondary transmission in schools, despite its notable decrease after intervention implementation. This unexpected finding might be explained by two factors. First, before the intervention, most classes were put in quarantine immediately independently of the presence of secondary transmission in the class, considering all classmates as close contacts, thus, delay in testing was not relevant for secondary transmission in these classes. Furthermore, the unmeasured tracing delay in the family/community better reflects the intervention efficacy and represents the timeliness in linking SARS-CoV-2 positive children to the school investigation. A better link between sporadic cases in households to school exposure after the

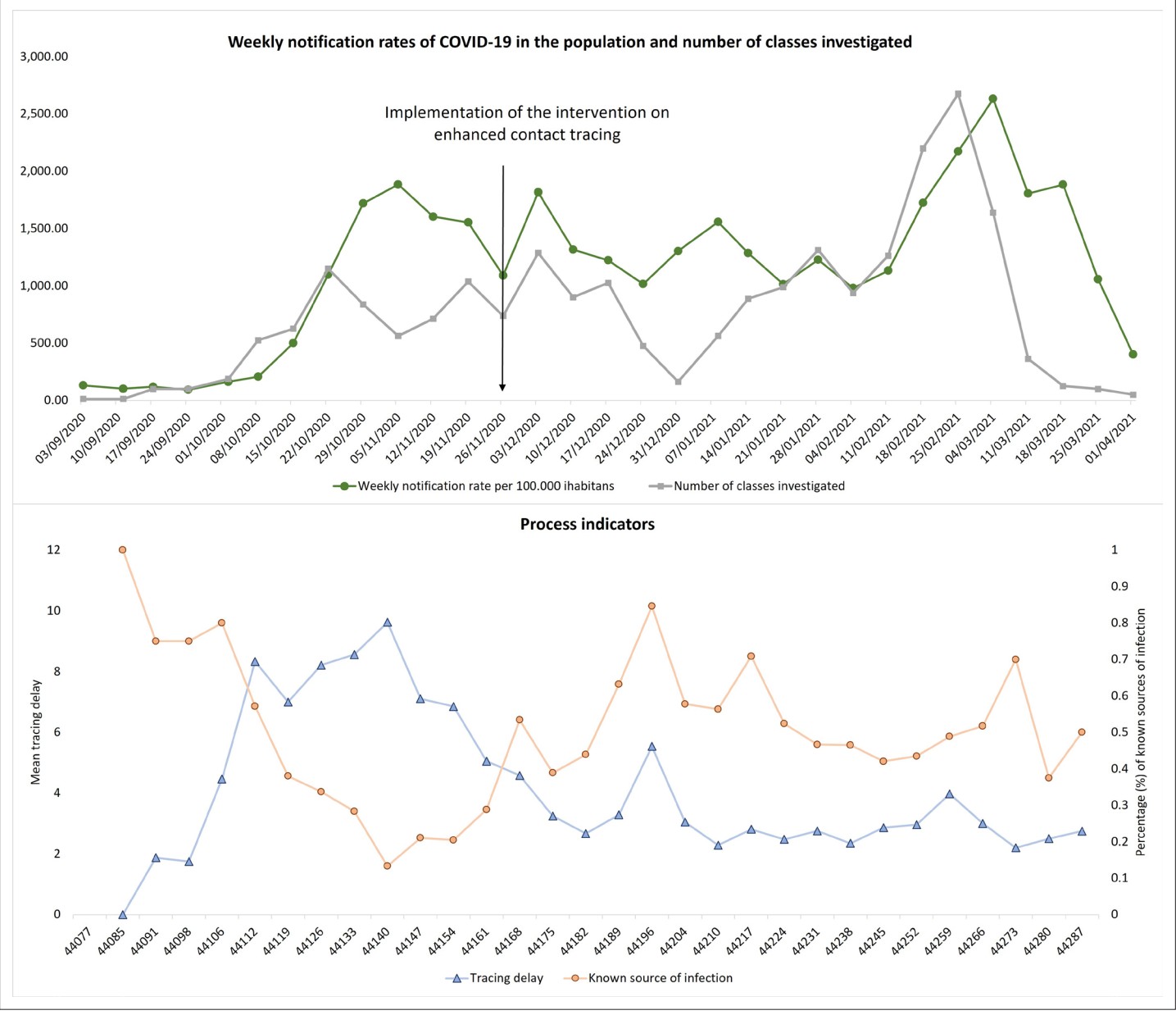

**Figure 5.** Upper graph: Weekly notification rates of COVID-19 per 100.000 inhabitants of Reggio Emilia Province, 1 September 2020 – 4 April 2021 and number of classes investigated. Lower graph: Weekly average contact tracing delay and percentage of index cases with a known source of infection.

**Table 4.** Negative binomial regression of the association between the number of secondary cases (outcome) and intervention promptness indicators (mediators).

| | Entire period (n=1884) | | Before Alpha variant (n=827) | |
|---|---|---|---|---|
| | IRR* | 95% CI | IRR* | 95% CI |
| Tracing delay | 1.01 | 0.99–1.04 | 1.03 | 0.99–1.07 |
| Known source of infection of the index case | 0.75 | 0.63–0.91 | 0.73 | 0.55–0.96 |
| Index case symptomatic | 1.21 | 0.96–1.53 | 1.30 | 0.93–1.82 |

*Adjusted for the type of school, type of index case, and class size.

**Table 5.** Four-way decomposition mediation analysis of the association between intervention and the number of secondary cases.

|  | ERR* | 95% CI |
|---|---|---|
| Total effect | –0.1 | –0.35–0.15 |
| Controlled direct effect |  |  |
| Known contact (M=1) | –0.29 | –0.61–0.03 |
| Unknown contact (M=0) | 0.31 | –0.49 to –0.02 |
| Pure indirect effect | –0.02 | –0.10–0.07 |
| Mediated interaction | –0.14 | –0.28 to –0.01 |
| Reference interaction |  |  |
| Known contact (M=1) | 0.35 | 0.03–0.68 |
| Unknown contact (M=0) | –0.25 | –0.49 to –0.02 |

*Adjusted for the type of school, type of index case, and class size.
ERR = excess relative risk. M = mediator (known source of infection of the index case).

intervention implementation is also supported by the higher fraction of asymptomatic index cases identified as well as the higher fraction of index cases that were part of a household cluster.

The direct effect of the intervention would lead to an almost 30% reduction in secondary transmission if the source of infection of all index cases was known. Moreover, the known source of infection had a greater impact on the secondary transmission when acting in the interaction with the intervention than independently (14% vs 2% reduction in the number of secondary cases).

The four-way decomposition analysis also showed that interaction alone accounted for a considerable part of the excess risk associated with the intervention. This practically means that knowing the source of infection of the index case in the period before the intervention, i.e., when contacts are not promptly tested, would have had a substantially detrimental effect on the secondary cases (35% increase). This possibly reflects that, before the intervention, often the source of infection for the school index case was identified during the field investigation and not before, thus, in the absence of BCT, knowing the source of infection is not a sign of timeliness at all.

The major limitation of the study is its before-and-after design; i.e., the impossibility to make an inference that observed changes are due to intervention and not due to other factors. In fact, multivariate and mediation analysis may not be enough to control for the fact that the force of infection was changing over the time series. It is often impossible to conduct properly designed experimental studies under a public health emergency. Nevertheless, it was impossible to apply our intervention to a limited number of schools, because it was an intervention targeting household clusters, in a particularly critical moment, i.e., during the peak of the second pandemic wave. The only way to assess the effectiveness of this intervention was to design an observational study trying to minimise the effect of confounding. A possible solution was testing the effect of mediators strictly linked to the intervention process (*Accorsi et al., 2021*). We adjusted analyses for major sources of confounding, but there are still unmeasured confounders. In fact, we could not classify the preventive measures put in place in each school, the time spent by each index case in the classroom, or the out-school contacts between classmates. Another important limitation is the lack of testing delay in a family/community as a process indicator, that we consider one of the real mechanisms of action of the new tracing strategy (first gray part of the conceptual scheme), but we assume this delay in the community follows the same trend as the delay observed in schools. Lastly, the same intervention may not yield the same results in a different epidemiological context, such as the presence of other variants of the virus (Omicron), or different control measures. However, it can have important public health implications in informing the management of the pandemic and the potential interaction between control measures in the family and in the school.

To our knowledge, this is the only study that attempted to quantify the potential effect of changing a contact tracing strategy in a community on secondary transmission in schools by estimating the excess risk associated with the intervention, through the application of a new mediation analysis method which allowed us to partition the total excess risk into separate effects of the intervention and its process indicators in the presence of their interaction (*VanderWeele, 2014*; *Discacciati et al., 2019*). As such it can have important methodological implications as well.

## Conclusion
Changing the contact tracing strategy in the community, from testing contacts at the end of quarantine to testing contacts immediately, reduced the time of contact tracing and increased the ability to

identify the source of infection in school outbreaks. The improvement in tracing performance appears to be linked to a decrease in the number of secondary cases in school contacts, although the intervention was implemented in a changing context just after the incidence peak of the autumn wave, and we cannot exclude that the observed differences are due to differences in the force of infection and to other control measures put in place before as the reduction of in presence school attendance.

## Acknowledgements

This project was carried out with the technical and financial support of the Italian Ministry of Health – CCM 2020 and Ricerca Corrente Annual Program 2023. Publication costs have been covered by the Emilia-Romagna Regional Health Authority, grant DGR591/2023.

## Additional information

### Funding

| Funder | Grant reference number | Author |
| --- | --- | --- |
| Ministero della Salute | CCM-2020 | Paolo Giorgi Rossi |
| Ministero della Salute | Ricerca Corrente Annual Program 2023 | Paolo Giorgi Rossi |

The funders had no role in study design, data collection and interpretation, or the decision to submit the work for publication.

### Author contributions

Olivera Djuric, Data curation, Formal analysis, Investigation, Visualization, Methodology, Writing - original draft; Elisabetta Larosa, Investigation, Methodology, Writing - review and editing; Mariateresa Cassinadri, Silvia Cilloni, Davide Pepe, Investigation; Eufemia Bisaccia, Patrizio Pezzotti, Conceptualization, Methodology; Laura Bonvicini, Formal analysis, Visualization; Massimo Vicentini, Francesco Venturelli, Investigation, Writing - review and editing; Paolo Giorgi Rossi, Conceptualization, Supervision, Funding acquisition, Methodology, Writing - review and editing; Alberto Mateo Urdiales, Formal analysis; Emanuela Bedeschi, Conceptualization, Supervision, Methodology; The Reggio Emilia Covid-19 Working Group, Data curation, Investigation

### Author ORCIDs

Olivera Djuric http://orcid.org/0000-0002-8574-5938
Francesco Venturelli http://orcid.org/0000-0002-9190-8668
Paolo Giorgi Rossi http://orcid.org/0000-0001-9703-2460

### Decision letter and Author response

Decision letter https://doi.org/10.7554/eLife.85802.sa1
Author response https://doi.org/10.7554/eLife.85802.sa2

## Additional files

### Supplementary files

• Supplementary file 1. Plain language definitions of the mediation analysis.
• Supplementary file 2. STATA code.
• Supplementary file 3. Weekly notification rates of new COVID-19 cases in Reggio Emilia.
• Supplementary file 4. Weekly average contact tracing delay and percentage of index cases with a known source of infection.
• MDAR checklist

## Data availability

According to Italian law, anonymized data can only be made publicly available if there is no potential for the reidentification of individuals (https://www.garanteprivacy.it). Thus, the data underlying this study are available on request to researchers who meet the criteria for access to confidential data. In order to obtain data, approval must be obtained from the Area Vasta Emilia Nord (AVEN) Ethics Committee, who would then authorize us to provide aggregated or anonymized data. Data access requests should be addressed to the Ethics Committee at CEReggioemilia@ausl.re.it as well as to the authors at the Epidemiology unit of AUSL-IRCCS of Reggio Emilia at info.epi@ausl.re.it, who are the data guardians. Code that we have used to analyse the data can be found in the Supplementary file 2. Excel sheet with numbers used to plot the graphs can be found in the Supplementary files 3 and 4.

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
