## [Editor Report]

This study provides a useful assessment of the effect of testing contacts of cases in school classes when identified, rather than at the end of quarantine, on both the number of secondary infections and other outcomes including tracing delay and identification of the possible source of infection. The authors find solid evidence that the intervention led to a decrease in tracing delay, an increase in the known source of infection of index cases and that knowing the source of the index case reduced secondary transmission. The novel application of the mediation analysis will be of interest to infectious disease epidemiologists and the findings will be of interest to those planning for future pandemics.

---

## [Decision Letter]

**Decision letter after peer review:**

Thank you for submitting your article "Effect of an enhanced public health contact tracing intervention on the secondary transmission of SARS-CoV-2 in educational settings: the four-way decomposition analysis" for consideration by *eLife*. Your article has been reviewed by 2 peer reviewers, and the evaluation has been overseen by a Reviewing Editor and Diane Harper as the Senior Editor.

Essential revisions:

1) Several aspects of the analysis are unclear and need to be more fully and transparently described.

2) There needs to be a more complete consideration of important potential confounders.

3) More background on the mediation analysis approach used is needed, including the motivation for it, underlying assumptions, and a more detailed explanation of the four components (CDE, PIE, IntRef, IntMed) as it is unlikely that many readers will be familiar with this approach.

4) Several aspects of figure 1 need to be improved, and the authors should consider Reviewer 1's suggestion of a labelled timeline plot.

5) There needs to be more clarity regarding Table 1: in particular, what do the RRs represent and how were they derived?

6) Please use consistent language for key concepts and minimise the use of abbreviations.

7) There are some additional key references that should be cited.

*Reviewer #1 (Recommendations for the authors):*

Figure 1 would benefit from a visual legend for the different infection, testing, and quarantine states. The bold lines for quarantined households/schools/workplaces are not particularly clear (could possibly benefit from being thicker and a different colour/linetype). The blue circle state is not described in the figure legend.

I believe a labelled timeline plot would tell the story better, e.g. see Hellewell et al. 2020 and Kretzschmar et al. 2020.

Line 225: Avoid the use of the term "exposures" when describing the regression models in this study, as this may be confused with an infection exposure – suggest "explanatory variables".

The Methods section does not describe how the results in Table 1 were produced, and the values cannot be reproduced. In particular, the RRs do not seem to be relative to the "ref" category but instead the pre-intervention risk. The headings are not particularly informative so it is hard to know what the table is describing.

The mediation analysis would benefit from a more detailed explanation of what the CDE, PIE, IntRef and IntMed components represent in simple terms in the methods and results.

*Reviewer #2 (Recommendations for the authors):*

Thank you for conducting this analysis and writing this piece, and for your important work during the COVID-19 pandemic. This is a valuable dataset to add to the collective understanding of in-school transmission of COVID-19.

In addition to some of the limitations above, I recommend simplifying the analysis of your data, and concentrating on clarifying and enhancing the assessment of the relationship between the intervention and primary outcome as the central tenet of your paper. I did not follow all aspects of your four-way decomposition mediation analysis, but I think redefining and refocusing your primary analyses and providing more descriptions of these methods might make these more complex models more intelligible. The dataset you have is rich in detail and size and should allow for multivariate models to account for many potential confounders.

I also recommend using consistent language to refer to key concepts (e.g. "identifying the source of infection for the index case") and avoiding the use of abbreviations (e.g. BCT for backward contact tracing), which might improve understanding among readers.

In terms of tables and figures, I was a little confused by Table 1, which includes a column of relative risks – I am unsure what these relative risks are referring to (the total study population, or the 'before intervention' group, or the 'after intervention' group?), and what value they add. I do think Figure 2 is important to visually display the descriptive data on tracing/testing delay and known source of infection, but it is currently quite busy – to improve clarity, you could try displaying the number of classes experiencing cases as a bar chart in the background, or, more preferably, a different chart with the same timeline in a panel format, showing the overall incidence in the province and the number of classes experiencing cases (so that one panel shows a graph of COVID-19 incidence overall and in schools, and the other shows your two process measures).

[Editors' note: further revisions were suggested prior to acceptance, as described below.]

Thank you for resubmitting your work entitled "Effect of an enhanced public health contact tracing intervention on the secondary transmission of SARS-CoV-2 in educational settings: the four-way decomposition analysis" for further consideration by *eLife*. Your revised article has been evaluated by Diane Harper (Senior Editor) and a Reviewing Editor.

The manuscript has been substantially improved but there are some remaining issues that need to be addressed, as outlined below:

1. The abstract should (briefly) indicate the methods used which, to our knowledge, are new to the COVID-19 literature (currently it just says "we analysed").

2. The abstract should also state the overall aim more clearly. Currently, it says "We assessed the impact of testing contacts…" but doesn't say the impact on what. Can the text on lines 203-4 be used here "To test the hypothesis that the contact tracing strategy in the community had an impact on the secondary case transmission in schools", or a variation on this?

3. The conclusions could be more nuanced than saying "seems to reduce" and if anything this doesn't give full credit to the contribution this paper makes to the literature. It would be fair to say that the study provides evidence of this, and there are strong reasons for thinking it is a causal link, though of course (as with all observational studies) it cannot definitively exclude other explanations.

4. Lines 70-74: As written suggests that these are current CDC and ECDC recommendations. This should be rewritten to clarify the time period these recommendations apply to.

5. Lines 110-111: It would be more appropriate to discuss the limitations in the discussion. It is also questionable referring to this as a "before-after study design" as that term is usually reserved for designs when a single observation is made before and after an intervention (though the term is not always used consistently). Note that, by itself, a before-after study is a very weak study design that would not usually be considered to provide useful causal information. This Cochrane guide to study designs may be useful here though this study may not fit well into any of these classifications.

6. Line 256 "The number of symptomatic index cases significantly decreased ". Unclear if "significantly" is supposed to imply this was clinically or statistically significant, and the authors should note that if the latter meaning is intended dichotomisation of results into statistically significant or otherwise is widely discouraged (see, for example, the American Statistical Association's statement on p-values https://www.amstat.org/asa/files/pdfs/p-valuestatement.pdf ). I would suggest deleting the word "significantly", and if the intention is to highlight the degree of certainty about this decline it would be better to report confidence intervals. See also lines 272, 276, 306, 316, and 351 where similar concerns apply.

7. Line 264 "Secondary transmission was associated only with type of index case" Though note that there was some (albeit weaker) evidence for reduced transmission in middle schools.

---

## [Author Response]

Essential revisions:1) Several aspects of the analysis are unclear and need to be more fully and transparently described.

To better describe the analyses we have made the following changes:

we have better described the primary outcome used in the mediation analysis and we have explained the other descriptive measures reported to describe the investigations;we have added a plain language explanations of the mediation analysis outputs;we have completely changed the Table 1 that was the main sources of confusion about outcomes and models.

2) There needs to be a more complete consideration of important potential confounders.

Our models are already adjusted for the main confounders we could measure: class size, type of school (including age of students), and type of index case (teacher or student). We have better described this in the new version of manuscript. Unfortunately, we cannot control for other and more important confounders, such as changes in children and family behaviors, changes in the epidemic spread etc. The four-way mediation analysis, allowing for interaction between period/intervention and mediators, may help interpreting the results also in the light of changes occurred in the environment concomitantly with the intervention. Furthermore, we limited the analysis to the period to the wild type period excluding the α variant period. We have better discussed this limitation of the study reasoning about their possible impact on the results.

3) More background on the mediation analysis approach used is needed, including the motivation for it, underlying assumptions, and a more detailed explanation of the four components (CDE, PIE, IntRef, IntMed) as it is unlikely that many readers will be familiar with this approach.

We added a paragraph to explain better the mediation analysis, including a supplementary box in which all the outputs are defined in plain language.

4) Several aspects of figure 1 need to be improved, and the authors should consider Reviewer 1's suggestion of a labelled timeline plot.

We have made a new figure 1.

5) There needs to be more clarity regarding Table 1: in particular, what do the RRs represent and how were they derived?

Table 1 has been completely changed. We have made a descriptive table 1 showing the characteristics of the epidemics before and after the intervention. Then we have added a table showing the associations between the class or index case characteristics and the outcome in order to show the possible confounders.

6) Please use consistent language for key concepts and minimise the use of abbreviations.

We have limited the abbreviations to few and very common cases.

7) There are some additional key references that should be cited.

We have updated and integrated the references.

Reviewer #1 (Recommendations for the authors):Figure 1 would benefit from a visual legend for the different infection, testing, and quarantine states. The bold lines for quarantined households/schools/workplaces are not particularly clear (could possibly benefit from being thicker and a different colour/linetype). The blue circle state is not described in the figure legend.I believe a labelled timeline plot would tell the story better, e.g. see Hellewell et al. 2020 and Kretzschmar et al. 2020.

We thank the reviewer for this suggestion. We prepared a new figure with timeline plot consulting the papers suggested. We also improved previous figure 1.

Line 225: Avoid the use of the term "exposures" when describing the regression models in this study, as this may be confused with an infection exposure – suggest "explanatory variables".

We have replaced “exposure” with “intervention”.

The Methods section does not describe how the results in Table 1 were produced, and the values cannot be reproduced. In particular, the RRs do not seem to be relative to the "ref" category but instead the pre-intervention risk. The headings are not particularly informative so it is hard to know what the table is describing.

We are sorry, the Table 1 was not correct and it was confusing. We have completely changed it and split in two tables.

The mediation analysis would benefit from a more detailed explanation of what the CDE, PIE, IntRef and IntMed components represent in simple terms in the methods and results.

We tried to better explain the meaning of the mediation analysis components.

Reviewer #2 (Recommendations for the authors):Thank you for conducting this analysis and writing this piece, and for your important work during the COVID-19 pandemic. This is a valuable dataset to add to the collective understanding of in-school transmission of COVID-19.In addition to some of the limitations above, I recommend simplifying the analysis of your data, and concentrating on clarifying and enhancing the assessment of the relationship between the intervention and primary outcome as the central tenet of your paper. I did not follow all aspects of your four-way decomposition mediation analysis, but I think redefining and refocusing your primary analyses and providing more descriptions of these methods might make these more complex models more intelligible. The dataset you have is rich in detail and size and should allow for multivariate models to account for many potential confounders.

We thank the reviewer for this suggestion. We completely re-framed the preliminary analyses to introduce step by step the mediation analysis. Probably this is not really a simplification; nevertheless, we hope it makes clearer the logical links between all the tables and the final synthesis made by the mediation analysis. In this logical flow, we also present the analysis of confounders that was not explicit in the previous version.

I also recommend using consistent language to refer to key concepts (e.g. "identifying the source of infection for the index case") and avoiding the use of abbreviations (e.g. BCT for backward contact tracing), which might improve understanding among readers.

We thank the reviewer and we tried to be consistent in the language and reduced the use of acronyms.

In terms of tables and figures, I was a little confused by Table 1, which includes a column of relative risks – I am unsure what these relative risks are referring to (the total study population, or the 'before intervention' group, or the 'after intervention' group?), and what value they add.

We thank the reviewer. We apologize; surely, Table 1 was confounding and wrong in its conception. We completely reviewed the table.

I do think Figure 2 is important to visually display the descriptive data on tracing/testing delay and known source of infection, but it is currently quite busy – to improve clarity, you could try displaying the number of classes experiencing cases as a bar chart in the background, or, more preferably, a different chart with the same timeline in a panel format, showing the overall incidence in the province and the number of classes experiencing cases (so that one panel shows a graph of COVID-19 incidence overall and in schools, and the other shows your two process measures).

We thank the reviewer. We accordingly changed the figure and put in two separate panels the incidence, the number of classes investigated, and the process indicators.

[Editors’ note: what follows is the authors’ response to the second round of review.]

The manuscript has been substantially improved but there are some remaining issues that need to be addressed, as outlined below:1. The abstract should (briefly) indicate the methods used which, to our knowledge, are new to the COVID-19 literature (currently it just says "we analysed").

We have added a sentence in the abstract that explains which type of analysis has been used and for which reason.

2. The abstract should also state the overall aim more clearly. Currently, it says "We assessed the impact of testing contacts…" but doesn't say the impact on what. Can the text on lines 203-4 be used here "To test the hypothesis that the contact tracing strategy in the community had an impact on the secondary case transmission in schools", or a variation on this?

Thank you, we have changed the background in the abstract as suggested.

3. The conclusions could be more nuanced than saying "seems to reduce" and if anything this doesn't give full credit to the contribution this paper makes to the literature. It would be fair to say that the study provides evidence of this, and there are strong reasons for thinking it is a causal link, though of course (as with all observational studies) it cannot definitively exclude other explanations.

Thank you, we have changed the conclusions in the abstract as suggested.

4. Lines 70-74: As written suggests that these are current CDC and ECDC recommendations. This should be rewritten to clarify the time period these recommendations apply to.

We have corrected this part of the introduction.

5. Lines 110-111: It would be more appropriate to discuss the limitations in the discussion. It is also questionable referring to this as a "before-after study design" as that term is usually reserved for designs when a single observation is made before and after an intervention (though the term is not always used consistently). Note that, by itself, a before-after study is a very weak study design that would not usually be considered to provide useful causal information. This Cochrane guide to study designs may be useful here though this study may not fit well into any of these classifications.

We agree with the Editor that a before-after study is a weak study design that does not allow to make causal inference. As explained (lines 198-203 and 225-242), we have applied a mediation analysis that allows the interaction between intervention (before and after the implementation of public health intervention) and mediator (process indicators) to overcome this methodological issue.

6. Line 256 "The number of symptomatic index cases significantly decreased ". Unclear if "significantly" is supposed to imply this was clinically or statistically significant, and the authors should note that if the latter meaning is intended dichotomisation of results into statistically significant or otherwise is widely discouraged (see, for example, the American Statistical Association's statement on p-values https://www.amstat.org/asa/files/pdfs/p-valuestatement.pdf ). I would suggest deleting the word "significantly", and if the intention is to highlight the degree of certainty about this decline it would be better to report confidence intervals. See also lines 272, 276, 306, 316, and 351 where similar concerns apply.

We agree with the Editor. We have deleted “significantly” troughout the text and we have added a sentence in the statistical analysis that states that we have based our judgement on 95% confidence intervals.

7. Line 264 "Secondary transmission was associated only with type of index case" Though note that there was some (albeit weaker) evidence for reduced transmission in middle schools.

We have deleted “only”.